# Detection of Colorectal Cancer and Advanced Adenoma by Liquid Biopsy (Decalib Study): The ddPCR Challenge

**DOI:** 10.3390/cancers12061482

**Published:** 2020-06-06

**Authors:** Audelaure Junca, Gaëlle Tachon, Camille Evrard, Claire Villalva, Eric Frouin, Lucie Karayan-Tapon, David Tougeron

**Affiliations:** 1University of Poitiers, 86000 Poitiers, France; audelaure.junca@yahoo.fr (A.J.); gaelle.tachon@chu-poitiers.fr (G.T.); camille.evrard@chu-poitiers.fr (C.E.); eric.frouin@chu-poitiers.fr (E.F.); lucie.karayan-tapon@chu-poitiers.fr (L.K.-T.); 2Department of Pathology, University Hospital of Poitiers, 86000 Poitiers, France; 3Department of Cancer Biology, University Hospital of Poitiers, 86000 Poitiers, France; claire.villalva-gregoire@chu-poitiers.fr; 4U1084, Institut national de la santé et de la recherche médicale (INSERM), 86000 Poitiers, France; 5Department of Medical Oncology, University Hospital of Poitiers, 86000 Poitiers, France; 6Department of Gastroenterology, University Hospital of Poitiers, 86000 Poitiers, France

**Keywords:** circulating tumor DNA, liquid biopsy, droplet digital PCR, colorectal cancer, fecal occult blood test

## Abstract

Background: In most countries, participation in colorectal cancer (CRC) screening programs with the immunological fecal occult blood test (iFOBT) is low. Mutations of RAS and BRAF occur early in colorectal carcinogenesis and “liquid biopsy” allows detection of mutated circulating tumor DNA (ctDNA). This prospective study aims to evaluate the performance of RAS and BRAF-mutated ctDNA in detecting CRC and advanced adenomas (AA). Methods: One hundred and thirty patients who underwent colonoscopy for suspicion of colorectal lesion were included and divided into four groups: 20 CRC, 39 AA, 31 non-advanced adenoma and/or hyperplastic polyp(s) (NAA) and 40 with no lesion. Mutated ctDNA was analyzed by droplet digital PCR. Results: ctDNA was detected in 45.0% of CRC, in 2.6% of AA and none of the NAA and “no-lesion” groups. All patients with stage II to IV mutated CRC had detectable ctDNA (*n* = 8/8). Among the mutated AA, only one patient had detectable ctDNA (4.3%), maybe due to limited technical sensitivity or to a low rate of ctDNA or even the absence ctDNA in plasma. Specificity and sensitivity of KRAS- and BRAF-mutated ctDNA for the detection of all CRC and AA were 100% and 16.9%, respectively. Conclusions: ctDNA had high sensitivity in detection of advanced mutated CRC but was unable to sensitively detect AA. ctDNA analysis was easy to perform and readily accepted by the population but requires combination with other circulating biomarkers before replacing iFOBT.

## 1. Introduction

Colorectal cancer (CRC) is the third most common cancer worldwide. It represents 9% of all cancers with 1,400,000 new cases and 700,000 deaths per year, and it is the fourth most common cause of cancer death. Prognosis of early CRC (stage I and II) is good with overall survival at five years of about 90% in contrast to 15% in metastatic CRC (mCRC). In colorectal carcinogenesis, there is a multistep process from non-advanced adenoma (NAA) to advanced adenomas (AA) and then to cancer and metastatic lesions. AAs are adenomas with a high risk of transformation to cancer including high-grade dysplasia, villous or tubulovillous histology and/or size ≥ 10 mm and require colonoscopy surveillance every three years [1]. AA and early CRC are usually asymptomatic, but frequently entail occult bleeding which may be detected by immunochemical fecal occult blood test (iFOBT). The sensitivity of iFOBT performed every two years is ≈80% for CRC diagnosis and ≈30% for AA diagnosis with a specificity of ≈90% [2,3]. Many countries recommend colorectal cancer screening by iFOBT every two years for average-risk asymptomatic populations aged 50 to 74 years [4]. However, only a small part of the population is tested (as an example, 30% of the French population), mostly due to low acceptability of the sampling (stool) related to misinformation of the target populations [5,6].

In colorectal carcinogenesis, distinct genetic and epigenetic alterations from NAA to CRC are well-known. Indeed, approximately 40% of CRCs harbor *KRAS* exons 2 or 3 mutations [7] and 12% *BRAF V600E* mutations [8]. *KRAS* mutations are also observed in 10% of NAA and 30% to 60% of AA [9,10]. BRAF mutations are mostly reported in serrated polyps and present in 60% to 70% of them [11].

For many years, a high level of circulating free DNA (cfDNA) in blood has been described in cancer patients [12,13]. cfDNA can originate not only from apoptotic or necrotic tumor cells of the primary tumor or metastases, but also from turn-over of non-malignant cells. More recently, circulating tumor DNA (ctDNA) has been described as the fraction of cfDNA originating from tumors that carries the same genetic alterations as those of the primary tumor [14]. *KRAS* mutations were the first genetic alterations used to identify ctDNA in CRC patients, with high sensitivity (85–90%) and specificity (95–100%) for *KRAS*-mutated mCRC [15]. While ctDNA analysis promises to set the genetic profiling of tumors by a simple blood test called “liquid biopsy”, replacing exploration of tissue samples, whether it would be efficient for CRC screening remains an outstanding question. Perrone et al. analyzed *KRAS*-mutated ctDNA for the detection of early CRC and adenoma in a series of 170 patients with positive iFOBT. The rate of positive *KRAS*-mutated ctDNA was low (3%) and increased up to only 14.2% considering only patients with *KRAS*-mutated CRC or AA [16]. However, the ctDNA was analyzed by a mutant-enriched PCR, a less sensitive technique than new techniques such as droplet digital PCR (ddPCR).

Detection of ctDNA in patients with advanced cancer is easier than in patients at an early stage of cancer since the level of ctDNA could be very low (<1% of all circulating DNA) [15,17]. With its high sensitivity, up to 0.01%, ddPCR is one of the techniques in the best position to detect ctDNA at early tumor stages and to be used as a blood screening test [18,19]. Blood-based screening tests could also improve participation rates in CRC screening as compared to fecal tests. The objectives of the prospective DECALIB study were to evaluate early detection of colon lesions using cfDNA values, the concordance between tissue and plasma detection of *KRAS* and *BRAF* mutations and also the possibility of early detection of CRC lesions using *KRAS*- or *BRAF*-mutant ctDNA.

## 2. Results

### 2.1. Study Population

One hundred and fifty-five patients were included in this study. It is worth noting that all patients accepted the study, and that none refused to participate. Colonoscopy was performed mainly for personal/family history of polyp(s) (24.5%), complex mucosectomy of previously identified polyp(s) (23.2%) and gastrointestinal symptoms (21.3%) (Appendix A). Twenty-five patients were excluded, mostly due to failure of blood collection or polyp retrieval failure (Figure 1).

Finally, 20 patients were in the “CRC” group, 39 in the “AA” group, 31 in the “NAA” group and 40 in the “no-lesion” group. Mean age was 68.1 years (range 29–92) and 63.8% of patients were male. The 20 CRCs were classified by stage: seven stage 0 (pTis), six stage II, four stage III and three stage IV.

Among the 39 patients with at least one AA (“AA” group), 52 AA, 38 NAA and 4 hyperplastic polyps (HP) were resected during the colonoscopy. Among the 31 patients in “NAA” group 58 lesions (43 NAA and 15 HP) were resected. Finally, among the 130 patients analyzed 187 colorectal lesions were analyzed.

### 2.2. Molecular Status of Colorectal Lesions

Among the 20 CRCs, 12 had a mutation (60%) in at least one gene, nine had a *KRAS* mutation (45%) and three had a *BRAF* mutation (15%) (Table 1). Four patients presented additional lesions, mostly *KRAS* and *BRAF* wild-type (WT) NAA (*n* = 11/15) (Appendix A).

Concerning the 39 patients in the “AA” group, 23 patients had at least one mutated AA, 15 had at least one *KRAS*-mutated lesion (38.5%) and 8 had at least one *BRAF*-mutated lesion (20.5%). No patient had AA with both *KRAS* and *BRAF* mutations. Among the 52 AA analyzed 34.6% were *KRAS*-mutated (*n* = 18/52) and 17.3% were *BRAF*-mutated (*n* = 9/52) (Table 2). Most of the NAAs observed in the “AA” group were *KRAS* and *BRAF* WT (*n* = 35/42) (Appendix A).

Among the 31 patients in the “NAA” group, six had at least one *KRAS*-mutated lesion (19.4%) and 7 had at least one *BRAF*-mutated lesion (22.6%). In this group, 58 polyps were analyzed (41 tubular adenomas with low-grade dysplasia, 15 HP and 2 sessile serrated adenoma/polyps (SSA/*p*) and 13.8% were *KRAS*-mutated (*n* = 8/58) while 20.7% were *BRAF*-mutated (*n* = 12/58) (Table 3). All serrated polyps (hyperplastic polyps or sessile serrated adenoma/polyps) were *KRAS* or *BRAF*-mutated, mostly the latter (70.6%, *n* = 12/17).

No colorectal lesion had both *KRAS* and *BRAF* mutations. Only one patient (number 28) had a concomitant *KRAS*-mutated lesion and a *BRAF*-mutated lesion (Appendix A).

### 2.3. Circulating Cell Free DNA

The two methods used to quantify cfDNA (ddPCR and QuantiFluor^®^) showed comparable results and a high level of concordance (r = 0.89, *p* < 0.01) (Appendix A).

Median cfDNA level (QuantiFluor^®^) was significantly higher in the “CRC” group (13.53 ng/mL) compared to the “no-lesion” group (9.04 ng/mL; *p* = 0.004), the NAA group (8.65 ng/mL; *p* = 0.005) and the AA group (10.03 ng/mL; *p* = 0.024) (Figure 2). There was no significant difference between the “AA” and “NAA” groups compared to the “no-lesion” group (*p* = 0.29 and *p* = 0.64, respectively). In the “CRC” group median cfDNA level varied according to the tumor stage, ranging from 5.92 ng/mL in stage 0 to 41.14 ng/mL in stage IV. As regards to diagnostic performance, using a threshold at 12 ng/mL successfully identified 65.0% (*n* = 13/20), 28.2% (*n* = 11/39) and 25.8% (*n* = 8/31) of cancers, AA and NAA, respectively. The sensitivity of cfDNA (using a threshold at 12 ng/mL) in the detection of CRC or AA was then 40.7% while the specificity was 77.5%. The threshold of 12 ng/mL was determined using ROC curve comparing cfDNA from “no-lesion” group with cfDNA from CRC group and applying the Youden Index [20].

### 2.4. Circulating Tumor DNA

*KRAS* or *BRAF*-mutated ctDNA was found in ten out of the 130 plasmas analyzed (7.7%), nine in the “CRC” group and one in the “AA” group. No ctDNA was detected in the “NAA” and “no-lesion” groups. Among the nine CRC patients with positive ctDNA, six were *KRAS*-mutated and three *BRAF*-mutated (Table 4). All of the mutations detected in plasma were also identified in the primary tumor. The fraction of ctDNA among cfDNA varied according to the tumor stage, ranging from 0.09% in stage 0 to 8.6% in stage IV. The rate of cfDNA and ctDNA did not vary according to the presence or not of vascular emboli.

Taking all CRCs into consideration, the sensitivity of *KRAS* or *BRAF*-mutated ctDNA to detect the CRC was 45% (*n* = 9/20). When *KRAS* and *BRAF* WT CRC were excluded, sensitivity reached 75% (*n* = 9/12). All *KRAS* or *BRAF*-mutated advanced CRCs (stage II to IV) were detected by ctDNA. All patients with positive *KRAS* or *BRAF*-mutated ctDNA presented the same mutation in their tumor with no false positive cases. All patients with CRC and positive ctDNA who underwent curative surgery and were alive at one month after surgery undertook a second blood test (*n* = 5). No ctDNA was detected in their plasmas.

Concerning AA, only one patient had positive ctDNA (*n* = 1/39). This patient (number 63) had a *KRAS*-mutated 4 cm tubular adenoma with low-grade dysplasia and *KRAS*-mutated ctDNA was identified in his plasma. Sensitivity in detection of AA was 2.6% (*n* = 1/39) and 4.3% for detection of a mutated AA (*n* = 1/23). There was no false positive ctDNA result.

Altogether, the sensitivity of ctDNA to detect AA and CRC was 16.9% (*n* = 10/59) and the specificity showed a value of 100%. The positive predictive value in detection of AA and CRC was 100% and the negative predictive value was 59.2%. A comparison with the performance of iFOBT is shown in Table 5.

## 3. Discussion

The DECALIB study is the first prospective study to evaluate the diagnostic performance of *KRAS*- and *BRAF*-mutated ctDNA using a highly sensitive technique, ddPCR, to detect both AA and CRC. cfDNA level was higher in CRC patients compared to patients with normal colonoscopy, but no difference was observed for patients with AA or NAA versus patients with normal colonoscopy. All *KRAS* or *BRAF*-mutated advanced CRCs (stage II to IV) were detected by ctDNA. ctDNA had low sensitivity to detect *KRAS*- or *BRAF*-mutated CRC at stage 0 (in situ) and *KRAS*- or *BRAF*-mutated AA, 25% and 4.3% respectively. No ctDNA was identified in patients with *KRAS* and *BRAF* WT colorectal lesions (specificity of 100%). Therefore, while feasible, it would be necessary to increase ctDNA sensitivity in the screening of pre-cancerous lesions and early stage CRC.

As previously published, we observed a higher level of cfDNA in CRC patients as compared to the normal population [15,21,22]. There also existed a correlation between cfDNA level and tumor stage, especially in stage IV CRC, which presented a high level of cfDNA. The level of cfDNA in CRC patients varies according to the studies and techniques used, ranging from 10 to 5000 ng/mL [23,24]. Direct comparison between cfDNA levels across studies is difficult as no standardization is applied. In future studies, the different techniques applied shall need to be harmonized. In addition, as shown by the low specificity (77.5%) in our study, high cfDNA level is not tumor-specific and can also increase in a number of other diseases including physiological stress response, sepsis and inflammatory disease [25]. Concerning patients with AA or NAA, we did not notice any difference in cfDNA levels compared to patients with no colorectal lesion. It is worth noting that, to our knowledge, we are the first to examine the relevance of cfDNA level in patients with adenoma.

In contrast to cfDNA, ctDNA originates only from tumor tissue and shares the same mutational landscape. We consequently analyzed *KRAS* and *BRAF* mutations, which are well-known oncogenic mutations in CRCs. As they occur in approximately 50% of CRCs, our ddPCR test was able to detect at most half of the tumors. In the DECALIB prospective study, we evaluated both mutational status of plasma (*n* = 130) and all colorectal lesions (*n* = 187) in 130 patients. *KRAS* mutations were observed in 45.0% of CRCs, 34.6% of AAs and 13.8% of NAAs. *BRAF* mutations were less frequent in CRC (15.0%) but as expected, most serrated lesions had *BRAF* mutations (91.6% of SSA/*p* and 66.6% of HP). These mutational rates were in accordance with the literature [7,8,11].

One major limit of liquid biopsy is the detection of only advanced *BRAF-* or *KRAS*-mutated CRC but not *BRAF* and *KRAS* wild-type CRC and adenomas. Comparing ctDNA and tissue analyses, we found that the sensitivity and specificity of *KRAS*- and *BRAF*-mutated ctDNA in the detection of AA and CRC were 16.9% and 100%, respectively. Nevertheless, the sensitivity of ctDNA in the detection of either mutated CRC or mutated AA reached 28.6%. ctDNA was able to detect 75% of all mutated CRCs, 100% of advanced stages (stage II–IV) and 25% of early stages (stage 0). These results were comparable to those of the Dielh et al. study using BEAMing digital PCR [26]. They analyzed APC-mutated ctDNA in the plasma of 33 patients with *APC*-mutated colorectal lesions (11 AA and 22 CRC). *APC*-mutated ctDNA was detected in 51.5% of patients with 100% of detection in stage IV, 62.5% in stage I/II and 9.1% in AA. The fractional abundance of ctDNA varied according to tumor stage with an average of 0.04% for stage I and 11.1% for stage IV. The only patient with AA and detectable *APC*-mutated ctDNA had a ctDNA fractional abundance of 0.02%. Therefore, *APC* is not a satisfactory ctDNA biomarker and needs to be improved for stage I and AA detection. It is worth noting that, similar to our study, the number of patients with T1 classification was low (*n* = 1/22). We found similar rates too with a very low fractional abundance of *KRAS*- or *BRAF*-mutated ctDNA in the non-metastatic stage (<1%) as compared to the metastatic stage (8.6%).

In another study, Perrone et al. analyzed *KRAS*-mutated ctDNA by a mutant-enriched PCR as a means of detecting early CRC in a series of 170 iFOBT-positive subjects having undergone colonoscopy [16]. The cohort comprised 12 CRC, 19 AA, 73 NAA or HP and 63 patients with no lesion. The rate of *KRAS* mutations detected in plasma was very low (3%). In contrast to our study, colorectal lesions were not molecularly characterized for all patients; direct comparison with our study is therefore not possible. Kopreski et al., analyzed *KRAS*-mutated ctDNA in patients having undergone colonoscopy and were able to detect 35.5% of AAs and CRCs using a combination of PCR and restriction enzyme techniques [27]. However, they also detected *KRAS*-mutated ctDNA in 22% of patients with normal colonoscopy and presumed that this mutated ctDNA originated from aberrant crypts undetected by colonoscopy. Nevertheless, in our study, as in the other previously cited studies using sensitive techniques, we did not detect mutated ctDNA in the “no-lesion” group”; there may have been some false positives in the Kopreski et al. study due to mutant-enriched PCR method. Therefore, not only the biomarker but also the technique used is essential to efficiently screen CRC and AA. Finally, as regards CRC detection, other studies reported 20% to 100% success for mutated-CRC detection by ctDNA, in all instances with a lower rate of CRC detection in early stages [28,29]. It is worth noting that, unlike the other studies; ours was the only one to consider CRC at stage 0 and *BRAF*-mutated ctDNA. To sum up, notwithstanding the use of ddPCR, while we did not improve AA detection rate, we succeeded in detecting stage II to stage IV CRC and partially succeeded in detecting CRC at stage 0. Nevertheless, sensitivity of ctDNA remains lower than iFOBT for CRC detection. In addition, the cost of ctDNA analysis (≈30 euros) is higher than iFOBT (≈10 euros). The lack of detection of AA lesions could come from limited technical sensitivity, or from a low rate of ctDNA or even absence of ctDNA in the plasma of the patient. In these last cases, other biomarkers or better techniques would not change the result.

One way to improve the detection rate for colorectal lesions would be to analyze mutations other than *KRAS* and *BRAF*, such as *APC*, *PIK3CA*, *TP53* or epigenetic abnormalities such as an increased promoter methylation of *SEPTIN 9* (m*SEPT9*) [26,29,30,31]. mSEPT9 has been identified in almost 80% of CRCs. The Prospective Evaluation of Septin 9 (PRESEPT) study analyzed the plasma of 7941 patients who underwent screening colonoscopy using the commercial kit based on m*SEPT9* detection (Epi proColon^®^) and found sensitivity of 48.2% in detection of CRC, depending on tumor stage (35.0% in stage I, 63.0% in stage II, 46.0% in stage III and 77.4% in stage IV) [30]. Sensitivity and specificity in detection of AA were 11.2% and 88.4%, respectively. The rate of positive subjects with no colorectal lesion reached 9.1%. Despite that m*SEPT9* is present in 80% of all CRCs, the test based on m*SEPT9* detection was only able to detect half of the CRCs, with practically the same sensitivity as ours (45%), whereas we had no false positive subjects. Therefore, *mSEPT9* by itself seemed an unsatisfactory screening biomarker used alone and it is reasonable to think that a combination of m*SETP9*, *KRAS* and *BRAF* may increase CRC detection by ctDNA, but not enough for AA detection. Next-generation sequencing allows detecting multiple mutations simultaneously but has lower sensitivity than ddPCR (1% versus 0.01%). Some studies have reported NGS tests with high sensitivities in the range of 0.1% [32,33]. These techniques required deep sequencing and molecular barcoding strategy, which increase the overall complexity and cost, and are therefore not suitable for routine practice. In the near future, we could use other techniques with better analytical sensitivity to more efficiently detect pre-cancerous lesions by ctDNA, however, as is the case for ddPCR test, a low quantity of ctDNA will always be a limiting factor, as this quantity is known to increase with tumor size and invasiveness [15].

One limit of our study was the absence of stage I CRC. All eligible patients were included in this study but by fortuity there was no stage I. However, contrary to most previous studies, we included patients with in situ tumor (Stage 0) and successfully detected ctDNA for 25% of them. All *KRAS* or *BRAF*-mutated stage II CRCs were detected by ctDNA. Therefore, it is reasonable to think that most stage I CRC would be detected by ctDNA (between 25% and 100%).

Our study is of major interest for countries with low iFOBT participation rate. For countries with up to 60% iFOBT participation rate, the search for an alternative screening solution seems less needed [34]. However, ctDNA is an expanding field of research and analysis of ctDNA may become a common test for screening/detection of various tumors or pre-cancerous lesions in only one blood sample. Nevertheless, ctDNA currently lacks sensitivity to detect AA and our study needs to be considered as a negative study insofar as ctDNA cannot presently replace iFOBT. Despite the lack of sensitivity to detect early stage CRC and AA, ctDNA could already be used in addition to positive iFOBT since in many countries it takes a long time to program a colonoscopy. Indeed, if ctDNA is positive, colonoscopy should be performed as soon as possible since patients may have an advanced CRC due to the high positive predictive value of ctDNA to detect advanced CRC.

## 4. Materials and Methods

### 4.1. Patient Selection

Inclusion criteria were patients older than 18 years admitted for a total colonoscopy for suspicion of AA or CRC, including personal or family history of polyp(s), gastrointestinal bleeding, positive iFOBT and other symptoms suggestive of a digestive tract lesion (Appendix A). Both outpatients and inpatients were included. Non-inclusion criteria were patients with other malignancies within five years prior to study enrolment or not able to comply with the protocol procedures. Exclusion criteria also comprised incomplete colonoscopy, failure of blood collection and patients with no pathological analysis of resected lesions. From November 2015 to October 2016, 155 patients who underwent a total colonoscopy for suspicion of AA or CRC were included after signature of written consent. Protocol was approved by the French “Committee for the Protection of Persons West III” (DC 2015–2449) and was registered at the “National Commission on Informatics and Liberty” (10/02/2020).

### 4.2. Pathological Results

During colonoscopy all samples collected (resection or biopsies) underwent routine pathological analyses but also molecular analyses (*KRAS* and *BRAF*). Patients were divided into 4 groups: patients with CRC (“CRC” group), patients with at least one AA (“AA” group), patients with non-advanced adenoma(s) and/or hyperplastic polyp(s) (“NAA” group), and patients with no colorectal lesion (“no-lesion” group). Pathological examinations were performed by expert gastrointestinal pathologists from the University Hospital of Poitiers, blinded to the ctDNA results. AAs were defined as adenomas with high-grade dysplasia, villous or tubulovillous histology and/or size ≥ 10 mm. Sessile serrated adenoma/polyps (SSA/*p*) and traditional serrated adenomas (TSA) with dysplasia and/or ≥10 mm were also considered as AA. In the case of multiple synchronous lesions, patients were classified in the most severe group. CRC were classified using the American Joint Committee on Cancer (Issue 7, 2010). For patients with curative surgery for CRC and positive ctDNA, a second blood sample was collected one month afterwards.

### 4.3. Blood Samples, Colorectal Lesions and DNA Extraction

Blood samples were collected just before the colonoscopy, stored at 4 °C and processed within 1–4 h. Because of ethical concerns, the blood sample had to be taken while taking others samples for routine purpose. Blood samples were centrifuged at 3000 rpm for 10 min at 4 °C and plasma samples were stored at −80 °C. Prior to DNA extraction, 5 mL of plasma were centrifuged at 13,200 rpm for 10 min at 4 °C. DNA were extracted using the QIAamp Circulating Nucleic Acid kit^®^ (Qiagen, Hilden, Germany) according to the manufacturer’s instructions.

Quantification of cfDNA was evaluated by two methods: QuantiFluor^®^ dsDNA System (Promega, Madison, WI, USA) and QX200 Droplet Digital PCR system^®^ (Bio-Rad, Hercules, CA, USA). Positive (standard DNA at 100 ng/μL) and negative (H2O) internal quality controls were systematically used to calibrate the fluorometer.

Formalin-fixed, paraffin-embedded (FFPE) colorectal lesions were evaluated and macrodissected by expert pathologists. DNA extraction was performed using KAPA Express Extract Kits^®^ (KAPA Biosystems, CliniSciences, Nanterre, France).

### 4.4. Molecular Analyses

For each patient the same volume of the eluted DNA (9 µL) was used to determine *KRAS* and *BRAF* mutational status in plasma. Multiplexed probes for simultaneous investigation of seven mutations (p.G12A, p.G12C, p.G12D, p.G12R, p.G12S, p.G12V and p.G13D) of *KRAS* were used for ddPCR were purchased from (Bio-Rad). The p.V600E mutation of *BRAF* was not part of the multiplex and was investigated separately (Bio-rad). Mutated probes were linked to FAM (fluorescein) dye, and wild-type (WT) probes to HEX (hexachloro-fluorescein) dye. Genomic DNA from *KRAS/BRAF* WT and mutated cell lines were used as negative and positive controls. ctDNA analyses were blind to the colonoscopy results. ddPCR has been already implemented in our laboratory for routine purpose.

Analyses were conducted using the QX200 ddPCR system^®^ (Bio-Rad) according to manufacturer’s instructions. Briefly, 20 µL reaction mix containing specific probes, primers and cfDNA were set into the Droplet Generator Cartridge^®^ with suitable volume of QX200 Droplet Generator Oil^®^ for droplet generation (Bio-Rad). PCR amplification was performed on a C1000 Touch™ Thermocycler^®^ (Bio-Rad). The 96-well PCR plate was loaded into the QX200 Droplet Reader^®^ for fluorescence analyses and data were interpreted using QuantaSoft™ software version 1.7.4.0917 (Bio-Rad). The limit of detection (LOD) according to the supplier was 0.1% for the *BRAF* mutation assay and 0.2% for the *KRAS* multiplex assay. LOD has been assessedon site down to 0.05% by serial dilution of DNA from mutated cell lines. Only the droplets in the positive area for FAM signal were taken into account in mutation detection (Appendix A) [35].

DNA extracted from FFPE colorectal lesions were analyzed using pyrosequencing with Q24 PyroMark system^®^ (Qiagen, Hilden, Germany). CE-IVD allele-specific (Qiagen) or homemade primers were used to selectively amplify and detect *KRAS* and *BRAF* wild-type or mutant alleles, as previously described [36]. The same seven *KRAS* mutations and V600E mutation of *BRAF* were investigated in the plasma and the tissue.

### 4.5. Statistical Analysis

The primary endpoint of our study was to evaluate the diagnostic performance of ctDNA (sensitivity, specificity, positive and negative predictive values) in detection of CRC and AA. The secondary endpoint was to evaluate the performance of cfDNA in detection of CRC and AA. Rates of cfDNA obtained by QuantiFluor^®^ and ddPCR tests were compared using the Spearman correlation test. Rates of the circulating DNA in each group (“CRC”, “AA” and “NAA” groups) were compared to the “no-lesion” group using the Mann-Whitney test.

All statistical analyses were done using Statview 4.0 software (SAS Institute Inc., Cary, NC, USA) with a significance level of 0.05.

## 5. Conclusions

To our knowledge, our prospective DECALIB study was the first to use ddPCR for CRC and AA detection [37]. Our study showed that ddPCR was sensitive and specific for the detection of advanced CRC, but not for in situ carcinomas or advanced adenomas. Though our test remains to be improved and validated in a larger prospective trial, we have demonstrated that blood tests using ddPCR for CRC and AA detection is easy to perform and probably accepted by the population. Knowing that rates of AA detection by iFOBT are low and that participation rates barely reach 30% of the population, herein we present an alternative screening model to address the urgent need for successful colorectal lesion detection.

## Figures and Tables

**Figure 1 cancers-12-01482-f001:**
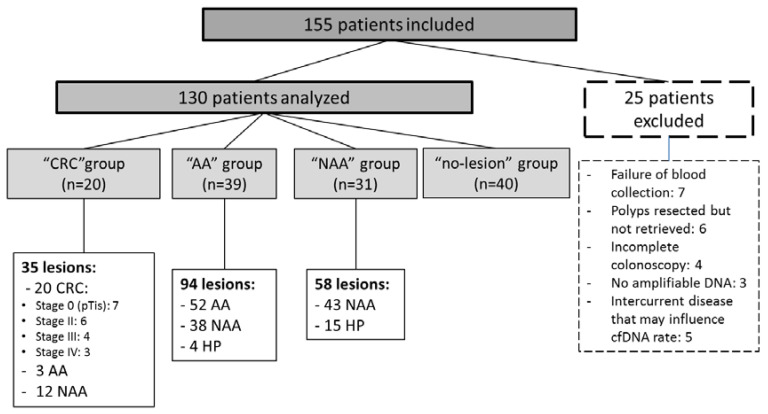
Flowchart. CRC group: colorectal cancer, AA group: patients with at least one advanced adenoma, NAA group: patients with non-advanced adenoma(s) and/or hyperplastic polyp(s), no-lesion group: patients with no colorectal lesion, cfDNA: circulating free DNA, AA: advanced adenoma, NAA: non-advanced adenoma, HP: hyperplastic polyps.

**Figure 2 cancers-12-01482-f002:**
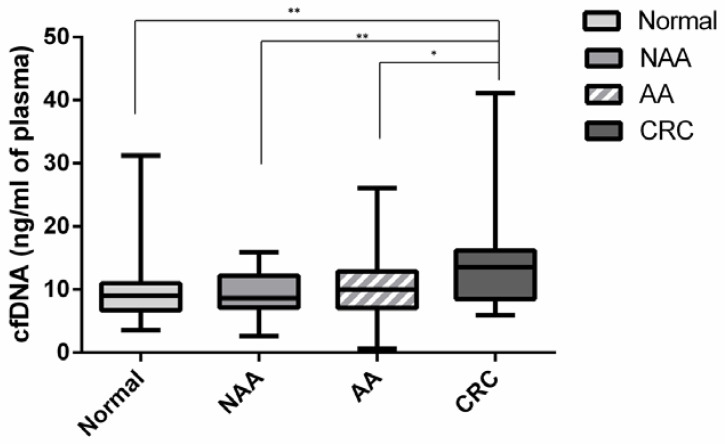
Box plots for the distribution of circulating free DNA concentrations in each group by QuantiFluor^®^. Each box indicates the 25th and 75th percentiles. The horizontal line inside the box indicates the median and the whiskers the extreme measured values. The Mann–Whitney test was performed for determination of statistical significance. * *p* < 0.05; ** *p* < 0.01. Normal: no lesion group; NAA: non-advanced adenoma group; AA: advanced adenoma group; CRC: colorectal cancer group.

**Table 1 cancers-12-01482-t001:** Molecular status of colorectal cancers according to the tumor stage.

Molecular Status	Stage					
	0	I	II	III	IV	Total (%)
*KRAS* mutation	4	-	3	1	1	9 (45)
*BRAF* mutation	-	-	1	2	-	3 (15)
*KRAS* and *BRAF* WT	3	-	2	1	2	8 (40)
Total (%)	7 (35)	-	6 (30)	4 (20)	3 (15)	20 (100)

WT: wild-type.

**Table 2 cancers-12-01482-t002:** Molecular status of advanced adenomas according to the pathological subtype.

Molecular Status	Pathologic Subtype					
	TALG ≥ 1 cm	TAHG	TVALG	TVAHG	SSA/*p* ≥ 1 cm or with dysplasia	Total (%)
*KRAS* mutation	5	4	8	1	-	18 (34.6)
*BRAF* mutation	-	-	-	-	9	9 (17.3)
*KRAS* and *BRAF* WT	13	1	9	1	1	25 (48.1)
Total (%)	18 (34.6)	5 (9.8)	17 (32.6)	2 (3.8)	10 (19.2)	52 (100)

TALG: tubular adenoma with low-grade dysplasia; TVALG: tubulovillous adenoma with low-grade dysplasia; TAHG: tubular adenoma with high-grade dysplasia; TVAHG: tubulovillous adenoma with high-grade dysplasia; SSA/*p*: sessile serrated adenoma/polyps; cm: centimeter; WT: wild-type.

**Table 3 cancers-12-01482-t003:** Molecular status of non-advanced adenomas according to the pathological subtype.

Molecular Status	Pathologic Subtype			
	TALG < 1 cm	SSA/*p* < 1 cm with no dysplasia	HP	Total (%)
*KRAS* mutation	3	-	5	8 (13.8)
*BRAF* mutation	-	2	10	12 (22.4)
*KRAS* and *BRAF* WT	38	-	-	38 (67.2)
Total (%)	41 (70.7)	2 (3.4)	15 (25.9)	58 (100)

TALG: tubular adenoma with low-grade dysplasia; SSA/*p*: sessile serrated adenoma/polyps; HP: hyperplastic polyp; WT: wild-type.

**Table 4 cancers-12-01482-t004:** Primary tumor and plasma status of “CRC” group.

Tumor Stage	pTNM *	Vascular Emboli	Mutation Identified in Primary Tumor	cfDNA (ng/mL)	Fraction of *KRAS*-Mutated ctDNA (%)	Fraction of *BRAF*-Mutated ctDNA (%)
**0**	pTis	-	***KRAS*** **G12C**	12.21	0	0
	pTis	-	WT	12.61	0	0
	pTis	-	***KRAS*** **G12D**	14.30	0.09	0
	pTis	-	***KRAS*** **G13D**	5.92	0	0
	pTis	-	WT	8.69	0	0
	pTis	-	WT	16.02	0	0
	pTis	-	***KRAS*** **G12V**	8.48	0	0
**II**	pT3	-	***BRAF*** **V600E**	15.09	0	0.58
	pT3	+	***KRAS*** **G13D**	9.87	0.25	0
	pT3	+	***KRAS*** **G12D**	25.58	0.08	0
	pT3	-	***KRAS*** **G12D**	16.17	0.15	0
	pT3	-	WT	29.04	0	0
	pT3	+	WT	8.33	0	0
**III**	pT4aN1b	-	***BRAF*** **V600E**	18.07	0	0.06
	pT3N1b	+	***BRAF*** **V600E**	16.17	0	0.16
	cT3N1M0	+	WT	6.06	0	0
	cT3N2M0	+	***KRAS*** **G13D**	12.76	0.59	0
**IV**	cTxNxM1	+	***KRAS*** **G12D**	14.66	8.6	0
	cTxNxM1	+	WT	41.14	0	0
	pT2N0M1	+	WT	6.89	0	0

* pTNM classification of CRC according to the American Joint Committee on Cancer (AJCC, issue 7, 2010) cfDNA: circulating free DNA, ctDNA: circulating tumor DNA, WT: wild-type.

**Table 5 cancers-12-01482-t005:** Sensitivity and specificity of ctDNA and immunochemical fecal occult blood test in detection of colorectal lesions.

Colorectal Lesion	Decalib All Colorectal Lesions Sensitivity/Specificity	Decalib *KRAS*- or *BRAF*- Mutated Colorectal Lesions Sensitivity/Specificity	Immunochemical Fecal Occult Blood Test (OC Sensor^®^) * Sensitivity/Specificity
CRC	45.0%/100%	75.0%/100%	65–85%/90–95%
AA	2.6%/100%	4.3%/100%	25–35%/90–95%
CRC plus AA	16.9%/100%	28.6%/100%	33–48%/90–95%

* References [2,3]; CRC: colorectal cancer; AA: advanced adenoma.

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
