# Peer review of "Detection of Colorectal Cancer and Advanced Adenoma by Liquid Biopsy (Decalib Study): The ddPCR Challenge"

_cancers, 2020, doi:10.3390/cancers12061482_

Round 1

Reviewer 1 Report

1. In countries with high health screening rates using iFOBT, how should they understand the results of this research?

2. In abstract the authors concluded that ctDNA analysis was easy to perform and readily accepted by the population but requires combination with other circulating biomarkers before replacing iFOBT. However, it's hard to catch up their opinion regarding other circulating biomarkers in paper discussion.

3. How about data of Stage I CRC? The finding from this research should not be applied to Stage I CRC?

4. The patient symbol in the Tables should be removed.

5. It's difficult to read and understand well from their discussion, whether or not this liquid biopsy approach will be promising in KRAS / BRAF wild-type CRC patients.

Author Response

Reviewer1:

Comments and Suggestions for Authors

  1. In countries with high health screening rates using iFOBT, how should they understand the results of this research?

It has been reported that some countries show a high iFOBT participation rate, up to 60% (Navarro et al., 2017). It is true that in these countries the search for an alternative screening solution is less needed. However, ctDNA is an expanding field of research and analysis of ctDNA may become a common test for screening/detection of various tumors or pre-cancerous lesions in only one blood sample. This point has been added in the Discussion paragraph (see page 8, lines 268-272).

Navarro, M., Nicolas, A., Ferrandez, A., and Lanas, A. (2017). Colorectal cancer population screening programs worldwide in 2016: An update. World J Gastroenterol 23, 3632–3642.

  1. In abstract the authors concluded that ctDNA analysis was easy to perform and readily accepted by the population but requires combination with other circulating biomarkers before replacing iFOBT. However, it's hard to catch up their opinion regarding other circulating biomarkers in paper discussion.

In our discussion, we already highlighted other circulating biomarkers and their advantages and the limits of them according to previous studies, like APC, PIK3CA, TP53 and mSEPT9 (line 246 to 266). In our opinion none of these circulating biomarkers have enough sensitivity to replace iFOBT and this opinion has now been added at several places in the Discussion paragraph (lines 218-220, lines 236-237, line 257-258). Nevertheless, no studies have assessed the combination of KRAS and BRAF markers with other known markers such as mSEPT9. Knowing that mSEPT9 has been identified in almost 80% of CRCs and BRAF and KRAS in half of CRCs, we can suppose that their association would increase the sensitivity of the ctDNA for CRC (lines 256-259).      

  1. How about data of Stage I CRC? The finding from this research should not be applied to Stage I CRC?

One limit of our study was the absence of stage I CRC. All eligible patients were included in this study but by fortuity there was no stage I. However, contrary to most previous studies, we included patients with in situ tumor (Stage 0) and successfully detected ctDNA for 25% of them. All KRAS- or BRAF-mutated stage II CRCs were detected by ctDNA. Therefore, it is reasonable to think that stage I CRC would be detected by ctDNA (between 25% and 100%). This assumption has been added lines 267-271.

  1. The patient symbol in the Tables should be removed.

Patient symbols have been removed from Table 4 and replaced by numbers in supp. Table 2.

  1. It's difficult to read and understand well from their discussion, whether or not this liquid biopsy approach will be promising in KRAS / BRAF wild-type CRC patients.

In our study, digital PCR is a targeted technique that aims to detect specific point mutation. Positive ctDNA meant that either BRAF or KRAS mutation was detected. Our technique design was then unable to detect KRAS and BRAF wild-type CRC patients. Indeed, in the Discussion paragraph, we now clearly say that this liquid biopsy has for major limit to detect only advanced BRAF or KRAS-mutated CRC, i.e. half of stage II to IV CRCs but not BRAF and KRAS wild-type CRC or adenomas  (page 7 lines 206-207 and 272-280). Besides, with KRAS/BRAF mutated ctDNA being efficiently detected in patients with mutated CRC by ddPCR, this technique can also detect other point alterations such as APC, TP53 or mSEPT9, the major issue remains to identify relevant circulating biomarkers to detect adenoma with a high sensitivity and specificity, which is not presently the case.  

Reviewer 2 Report

The work presented illustrates a very relevant problem in public medicine and gastrointestinal oncology such as the  non-invasive early diagnosis of colorectal cancers or advanced adenomas with a high risk of neoplastic progression.

The introduction/background provides a brief overview of the specific literature on the topic. The presentation of the data is linear, even if some limitations arise from the study that are not clearly stated by the Authors.
The results part contains 3 different sections: the possibility of early detection of colon lesions using cfDNA values, the concordance between tissue and plasma detection, the possibility of early detection of CRC lesions using KRAS or BRAF mutant ctDNA. However, only the third aim is clearly listed in the introduction (lines 76-77). Notably (and correctly), in the introduction the authors declare that the aim is to detect CRC and AA in general, not only KRAS or BRAF mut lesions. With regards to this point, as the Authors point out in the results, the study does not provide a high sensitivity and is thus to be considered negative. For this reason, the conclusions of the study should be modified to clearly underline the results upon the premises in the introduction, in other words the Authors cannot claim that the technique presents a 75% sensitivity. Since the ctDNA detection present a very high specificity (and hence a high PPV) but a low sensitivity (and NPV), this test could not be proposed as an alternative to either iFBOT or colonoscopy as a screening method. A potential clinical placement, given its hight PPV, could be before the colonoscopy, in order to increase the level of suspicion toward a (KRAS/BRAF mut) CRC in case of a positive iFOBT, as in reference 16 (Perrone et al), even if an improvement in the detection by including other markers such as SEPT9 or APC mutations would be desirable. However, with regards to the patients whose KRAS/BRAF mutational status really impacts on therapy (stage IV patients), this technique (ddPCR) has similar reliability compared to other liquid biopsy techniques (such as qPCR based), as shown by several other groups (Vivancos et al, Sci Rep 2019 and Vitiello et al, Cancers 2019).
With respect to the novelty, this work has evidenced that even a very sensitive analitical methodology is not able to identify ctDNA belonging to advanced adenomas. The Authors may correctly speculate that this limitation would present even including other genes to the panel (APC or others), since it is purely a technical limit and not a matter of absence of mutant DNA in these preneoplastic lesions. The latter observation would increase the overall relevance of the paper.
Taken together, the results presented have a scientific relevance and are well described, but the Authors should present the study as negative in its main aim, with interesting discussion points regarding cfDNA values, tissue-plasma concordance and impossibility to detect AA or NAA.
Below are listed some major and minor issues that should be addressed by the Authors:

Major issues:
-The main limitation of this study consists in the lack of predictivity of the technique with regards to KRAS and BRAF wt cancers, that are completely missed by the liquid biopsy. For this reason the real sensitivity of this method is low (45%) when confronted to the real numbers of CRC or AA. Since the comparator is a technique that identifies all types of lesions with respect to their molecular background, the main sensitivity value to be taken in consideration is not the 75%, that only represents a technical feature without an added clinical value. Please, specify the value for the global sensitivity/specificity of the test for molecularly unselected (wt+mut) lesions, in order to allow for a better comparison with the iFOBT.
-How did you pick the value of 12ng/ml for the cfDNA? Did you use a ROC? Please explain this point.
-What do you mean for the sensitivity of cfDNA in the identification of CRC (line 133)? You consider positive for cfDNA every sample above a specific value or you consider positive every cfDNA+?
-The specificity value in RAS or BRAF mut cancers does not correspond to the definition you give in lines 156-157. This value applies if you compare liquid biopsy with tissue KRAS/BRAF analysis, but not if you compare liquid biopsy with colonoscopy-based diagnosis of KRAS/BRAF mut lesions. In this case, specificity is TN/(TN+FP) , where TN are negative patients in both colonoscopy and liquid biopsy and FP are pts positive at ctDNA but not on colonoscopy. It might still be 100% in case of no false positive, but the definition should be adjusted.

Minor issues:
-Please specify what is intended with "no patient's refusal". Do you refer to no consent withdrawal or to the fact that all the patients to whom the study was proposed accepted it?
-patient exclusion: please explain what you intend with "failure of blood collection". I don't understand why a patient should be counted in the first place if the blood collection was not performed.
-In line 176: the values of sensitivity and specificity do not correspond to those previously stated.

Author Response

Reviewer 2

The work presented illustrates a very relevant problem in public medicine and gastrointestinal oncology such as the non-invasive early diagnosis of colorectal cancers or advanced adenomas with a high risk of neoplastic progression.

The introduction/background provides a brief overview of the specific literature on the topic. The presentation of the data is linear, even if some limitations arise from the study that are not clearly stated by the Authors.

We have now clarified the Discussion about the results of our study as suggested by both reviewers 1 and 2 (see page 7 and 8).

The results part contains 3 different sections: the possibility of early detection of colon lesions using cfDNA values, the concordance between tissue and plasma detection, the possibility of early detection of CRC lesions using KRAS or BRAF mutant ctDNA. However, only the third aim is clearly listed in the introduction (lines 76-77). Notably (and correctly), in the introduction the authors declare that the aim is to detect CRC and AA in general, not only KRAS or BRAF mut lesions.

We agree with this reviewer comment and now the 3 different aims have been mentioned in the Introduction (see page 2, lines 77-82).

With regards to this point, as the Authors point out in the results, the study does not provide a high sensitivity and is thus to be considered negative. For this reason, the conclusions of the study should be modified to clearly underline the results upon the premises in the introduction, in other words the Authors cannot claim that the technique presents 75% sensitivity.

We have now clarified the Discussion about the results of our study. We now clearly said that this liquid biopsy has sensitivity of 75% to detect BRAF or KRAS-mutated CRC but not adenoma and BRAF and KRAS wild-type CRC (see page 7, line 209-210). Indeed, as mentioned page 8, line 257-258, using BRAF and KRAS-mutated ctDNA detect only 45% of all CRC in our study. Moreover, we have modified the Discussion of the study (line 278-283).

Since the ctDNA detection present a very high specificity (and hence a high PPV) but a low sensitivity (and NPV), this test could not be proposed as an alternative to either iFBOT or colonoscopy as a screening method. A potential clinical placement, given its hight PPV, could be before the colonoscopy, in order to increase the level of suspicion toward a (KRAS/BRAF mut) CRC in case of a positive iFOBT, as in reference 16 (Perrone et al), even if an improvement in the detection by including other markers such as SEPT9 or APC mutations would be desirable. However, with regards to the patients whose KRAS/BRAF mutational status really impacts on therapy (stage IV patients), this technique (ddPCR) has similar reliability compared to other liquid biopsy techniques (such as qPCR based), as shown by several other groups (Vivancos et al, Sci Rep 2019 and Vitiello et al, Cancers 2019).

We totally agree that due to its insufficient sensitivity BRAF and KRAS-mutated ctDNA cannot actually replace iFOBT, especially for adenoma detection. As suggested, in case of positive iFOBT, ctDNA exploration could help to plan the colonoscopy, i.e. if ctDNA is positive, colonoscopy should be performed as soon as possible since patients may have an advanced CRC due to the high positive predictive value of ctDNA. This point is now discussed (page 8, lines 278-283, line 245-247 and lines 209-210). As mentioned page 8, line 258-261, it is reasonable to think that a combination of mSETP9, KRAS and BRAF and other point mutations like APC may increase CRC detection by ctDNA and perhaps will replace iFOBT in the future.

With respect to the novelty, this work has evidenced that even a very sensitive analitical methodology is not able to identify ctDNA belonging to advanced adenomas. The Authors may correctly speculate that this limitation would present even including other genes to the panel (APC or others), since it is purely a technical limit and not a matter of absence of mutant DNA in these preneoplastic lesions. The latter observation would increase the overall relevance of the paper. 

Taken together, the results presented have a scientific relevance and are well described, but the Authors should present the study as negative in its main aim, with interesting discussion points regarding cfDNA values, tissue-plasma concordance and impossibility to detect AA or NAA.

We don’t know if combination of multiple ctDNA biomarkers and higher sensitive techniques will improve adenoma detection in case of low rate ctDNA or no ctDNA. As suggested, we have modified the discussion of the manuscript to present as a rather negative study both in the Discussion paragraph and in the Abstract (see page 1 (lines 29-30) and 8 (lines 245-247 and 278-283)).

Below are listed some major and minor issues that should be addressed by the Authors:

Major issues:

-The main limitation of this study consists in the lack of predictivity of the technique with regards to KRAS and BRAF wt cancers, that are completely missed by the liquid biopsy. For this reason the real sensitivity of this method is low (45%) when confronted to the real numbers of CRC or AA. Since the comparator is a technique that identifies all types of lesions with respect to their molecular background, the main sensitivity value to be taken in consideration is not the 75%, that only represents a technical feature without an added clinical value. Please, specify the value for the global sensitivity/specificity of the test for molecularly unselected (wt+mut) lesions, in order to allow for a better comparison with the iFOBT. 

We highlighted the global sensitivity/specificity of the test in lines 170-171, Table 5 and lines 211-212, but we agree that this information wasn’t clear enough. Therefore, we also highlighted these values in the abstract line 31. Indeed, final result of the study is rather negative since KRAS- and BRAF- mutated ctDNA detects only 45% of all CRC and 4.3% of all adenoma and cannot, by itself, replace iFOBT (see page 1 (lines 29-30) and 8 (lines 245-247 and 278-283)).

-How did you pick the value of 12ng/ml for the cfDNA? Did you use a ROC? Please explain this point.

We used ROC curve comparing cfDNA from patient in the “no-lesion” group with cfDNA from patients in CRC group. To determine the cut-off of 12ng/ml we used the Youden Index (Youden, 1950).

This information has now been added in the manuscript lines 139-142.

Youden, W.J. (1950). Index for rating diagnostic tests. Cancer 3, 32–35.

-What do you mean for the sensitivity of cfDNA in the identification of CRC (line 133)? You consider positive for cfDNA every sample above a specific value or you consider positive every cfDNA+?

We have considered cfDNA as positive when level of cfDNA was above 12ng/µl and as negative when cfDNA was below 12ng/µl. We have changed the sentence to be clearer. 

-The specificity value in RAS or BRAF mut cancers does not correspond to the definition you give in lines 156-157. This value applies if you compare liquid biopsy with tissue KRAS/BRAF analysis, but not if you compare liquid biopsy with colonoscopy-based diagnosis of KRAS/BRAF mut lesions. In this case, specificity is TN/(TN+FP) , where TN are negative patients in both colonoscopy and liquid biopsy and FP are pts positive at ctDNA but not on colonoscopy. It might still be 100% in case of no false positive, but the definition should be adjusted.

We agree that we misused the word “specificity” in this sentence. We have consequently modified the sentence “All patients with positive KRAS- or BRAF-mutated ctDNA presented the same mutation in their tumor with no false positive case” (line 165).

Minor issues:

-Please specify what is intended with "no patient's refusal". Do you refer to no consent withdrawal or to the fact that all the patients to whom the study was proposed accepted it?

We referred to the fact the study was proposed and accepted by all patients, not a single one refused to participate to the study. The sentence has been changed accordingly lines 85-86.

-patient exclusion: please explain what you intend with "failure of blood collection". I don't understand why a patient should be counted in the first place if the blood collection was not performed.

These patients gave their consent for the study but blood sampling could not be performed due to clinical issue (organization, venipuncture failure…). Indeed, because of ethical concerns, blood samples had to be taken while taking other samples for routine purpose and not only for the purpose of the study.

This point has been added in the Methods lines 311-312.

In line 176: the values of sensitivity and specificity do not correspond to those previously stated.

The sensitivity cited line 176 concerned a specific group, the stage II to IV KRAS- or BRAF-mutated CRC. The sensitivity of this group was stated once previously in the text line 154-155 and the same information was given: “All KRAS or BRAF-mutated advanced CRCs (stage II to IV) were detected by ctDNA”. To prevent confusion for the readers we have modified the sentence by “All KRAS or BRAF-mutated advanced CRCs (stage II to IV) were detected by ctDNA” by contrast to provide specificity and sensitivity (lines 183-185).
